# Antibacterial, Antidiabetic, and Toxicity Effects of Two Brown Algae: *Sargassum buxifolium* and *Padina gymnospora*

Jesús Javier Alvarado-Sansininea [1], Rosario Tavera-Hernández [2], Manuel Jiménez-Estrada [2], Enrique Wenceslao Coronado-Aceves [3], Clara Inés Espitia-Pinzón [3], Sergio Díaz-Martínez [1], Lisandro Hernández-Anaya [1], Rosalva Rangel-Corona [4] and Alejandrina Graciela Avila-Ortiz [1,*]

[1] Herbario FEZA, Facultad de Estudios Superiores Zaragoza, Universidad Nacional Autónoma de México, Batalla de 5 de mayo S/N, Col. Ejército de Oriente, Mexico City 09230, Mexico

[2] Instituto de Química, Universidad Nacional Autónoma de México, Ciudad Universitaria, Coyoacán, Mexico City 04510, Mexico

[3] Instituto de Investigaciones Biomédicas, Universidad Nacional Autónoma de México, Mexico City 04510, Mexico

[4] Laboratorio de Oncología Celular, UMIEZ, Facultad de Estudios Superiores Zaragoza, Universidad Nacional Autónoma de México, Batalla de 5 de mayo S/N, Col. Ejército de Oriente, Mexico City 09230, Mexico

\* Correspondence: aviort27@gmail.com

**Abstract:** Seaweed has a variety or biological activities, including antibacterial, antioxidant, antidiabetic, and anti-inflammatory ones. Mexico has great macroalgae diversity, with nearly 1700 species; therefore, in this research two seaweeds from Mexico, *Sargassum buxifolium* and *Padina gymnospora*, were investigated for their antibacterial, antidiabetic, and toxic potential; and to understand their phytochemical components both were subjected to various extractions. Only the hexanic fraction was active, and the presence of fatty acids was detected. The two algal extracts showed interesting antimicrobial properties, which mostly inhibited the growth of Gram-positive bacteria (*E. faecalis*, *S aureus*, and *S. epidermidis*). The α-glucosidase activity was estimated for checking the antidiabetic capacity; *S. buxifolium* had best α-glucosidase inhibition compared with *P. gymnospora*. For toxicity, the hexanic extracts administered orally as nontoxic in the treated mice. These results suggest that the two algae have potential as resources for the development of antimicrobial agents.

**Keywords:** algae; antibacterial; Mexican; *Padina* and *Sargassum*

## 1. Introduction

The study of the phytochemical characteristics of marine natural resources like seaweeds is important, due to their role as an alternative source of new bioactive molecules. Seaweeds have been used since ancient times as food and as sources of medicinal drugs. The diversity of life conditions of seaweeds pushed them to develop many unique bioactive molecules [1,2], which exhibit antioxidant activity [3] and can be applicable for treatment of oxidative-damage related diseases or diabetes. Similarly, the use of marine natural products capable of bacteria inhibition [4] offers rich pharmacological potential. Many studies [5] have demonstrated the usefulness of seaweeds [6].

The coasts of Mexico hold great macroalgae diversity, having nearly 1700 species [7]. Of the total, more than 239 species belong to brown algae (class Phaeophyceae), although the actual number is still unknown and in constant flux due to the taxonomic work. The Gulf of Mexico is particularly rich in algal biodiversity [8], and some studies suggest great potential of organisms with pharmacological capabilities and as functional food [9]. Nevertheless, little is known about the antidiabetic and antibacterial effects of brown algae from de Gulf of Mexico. The genera *Sargassum* (Sargassaceae) and *Padina* (Dictyotaceae) have been taxonomically assessed in Mexico with morphological and molecular data to infer the

species limits [10]. *Sargassum* is a genus characterized by a branched thallus bearing blades resembling leaves and air bladders (in most of the species) that help with flotation. Nearly 16 species have been recognized in the Atlantic Coasts of Mexico, although the actual number of species is still under revision due to recent molecular results [10]. *Sargassum buxifolium* (Chauvin) M.J. Whynne is widely distributed in the Gulf of Mexico [11]. It is also reported to have a wide range of bioactive properties [12], making it an ideal subject for metabolites studies. On the other hand, *Padina* is a laminar fan-shaped thallus two or more layers of cells thick that usually grows in the intertidal zone, attached to rocky substrates. In Mexico, five species are distributed on the Pacific coast, and six are on the coast of the Gulf of Mexico and the Caribbean Sea, including *P. gymnospora* (Kützing) Sonder, a species characterized by having 6 to 8 cell layers [13]. The species is widely distributed, having been registered on the coast of Tamaulipas, Veracruz, Campeche, Yucatán and Quintana Roo states [14], and exhibits a variety of valuable medicinal properties, such as wound healing, antimicrobial, antidiabetic, and anti-inflammatory one [15–17]. Yet, its compounds have not been analyzed extensively.

As new emerging diseases and resistant strains of microorganism appear constantly anywhere, it is necessary to research for novel therapeutic compounds. According to the World Health Organization (WHO), antimicrobial resistance occurs when bacteria no longer respond to drugs, making treatment difficult, leading to infection, serious illness, and possibly death [18]. Therefore, the purpose of this research was to evaluate the biological activity—antioxidant, antidiabetic, and antibacterial—of extracts from *S. buxifolium* and *P. gymnospora* from the Gulf of Mexico.

## 2. Materials and Methods

### 2.1. Collection and Identification: Algae Material

Individuals of *S. buxifolium* and *P. gymnospora* were collected at Punta Delgada (19° 51′ 39″ N, 96° 27′ 36.5″ W) in the state of Veracruz, Mexico. The samples were identified according to their morphological characteristics and distribution. Reference vouchers were deposited in the FEZA Herbarium. Particularly, *P. gymnospora* samples correspond to the lineage 'E' sensu [14] which is supported by molecular taxonomic data.

### 2.2. Preparation of Extracts

Immediately after collection, both seaweeds were washed with sea water; epiphytes, associated organisms, sands, and other extraneous matter were removed. A total of 500 g of material was cut into small pieces and macerated. Both samples were extracted successively using hexane, ethyl acetate, methanol, and water, and were both macerated for 24 h at room temperature. The extracts were concentrated using distillation apparatus at 40 °C to obtain minimum quantity of crude extract.

### 2.3. NMR Analysis

The $^1$H-NMR spectra were recorded on a Brucker Avance III 400 MHz (Ettlingen, Germany) spectrometer using CDCl$_3$ as solvent. Residual solvent peaks were considered as a reference; displacement values are expressed in ppm.

### 2.4. Determination of the Fatty-Acid Profile by Fatty-Acid Methyl Esters (FAMEs) and Gas Chromatography (GC)

The fatty-acid (FA) profile was determined as fatty-acid methyl esters (FAMEs), which were prepared according to the following method: 1 mL of 0.5 M KOH was added to 16.2 mg of the hexanic extract of *S. buxifolium* and 21.3 mg of the hexane extract of *P. gymnospora*, and the mixture was boiled for 10 min. Subsequently, it was allowed to cool to room temperature, and 1 M HCl was added until pH 5. Extraction was carried out with hexane (2 × 1 mL), and 2 mL of methanol-boron trifluoride diethyl ether was added to the organic phase; this mixture was kept boiling for 10 min. The reaction mixture was allowed

to cool, and 1 mL of saturated NaCl solution was added. Extraction with hexane (2 × 1 mL) was performed, and the organic phase was dried over anhydrous sodium sulfate.

FAMEs were injected in duplicate to the gas chromatograph (Agilent 6890; Agilent, Santa Clara, USA) equipped with a flame ionization detector (FID) and AT-FAME column (30 m × 0.25 mm). The analytical conditions were: injection 1 μL, injector temperature 250 °C, detector temperature 250 °C. The temperature gradient in the column oven started at 180 °C for 15 min, followed by 10 °C/min increments up to 230 °C. The retention times of FAME standards were used to identify the chromatographic peaks of the samples. FA content was calculated based on the peak area.

*2.5. Biological Activity Assays*

2.5.1. Evaluation of the α-Glucosidase Inhibitory Activity

The assay was performed as previously reported [19] using an adapted method of Ye et al. 2010 [20]. α-Glucosidase type I (G 5003), *p*-nitrophenyl-α-D-glucopyranoside (N 1377, ≥99%), and quercetin (Q 0125, ≥98%) were purchased from Sigma Aldrich (Burlington, USA). Hexanic extracts were tested in triplicate at concentrations of 1, 10, and 100 μg/mL; and quercetin was used as positive control at 5 μg/mL.

2.5.2. Antibacterial Activity

Bacterial Strains

Bacterial strains used in this study: Gram-positive bacteria: *Enterococcus faecalis* American Type Culture Collection [ATCC] 51299, *Staphylococcus aureus* ATCC 25293, and *Staphylococcus epidermidis*; Gram-negative bacteria: *Escherichia coli* ATCC 25922, *Klebsiella pneumoniae*, *Pseudomonas aeruginosa* ATCC 10145, *Salmonella typhimurium*, *Escherichia coli* ESBL+ (resistant, extended spectrum beta lactamase) ATCC 700603, and *Klebsiella pneumoniae* ESBL+ obtained from the University of Sonora. Before testing, all bacterial strains were kept frozen at −70 °C in 10% glycerol broth.

Preparation of Working Solution

The extracts were dissolved in 100% dimethyl sulfoxide (DMSO, Sigma) (20 mg/mL) and kept at room temperature for 1 h to assure their sterilization [21]. These samples were diluted with fresh Mueller Hinton broth to its final concentrations of 3.125, 6.25, 12.5, 25, 50, 100, 200, and 400 μg/mL. The tested concentration of DMSO in all assays was 2% or less, which is nontoxic for bacteria.

Preparation of Inoculum

Bacterial colonies grown on Mueller Hinton agar (MCD Lab, Mexico State, Mexico) for 18–24 h (log phase of growth) were transferred to a sterile vial containing 15 mL of sterile 0.85% saline solution. The bacterial suspension was disaggregated by agitation using a Genie II vortex, speed 3, for 1 min, and left to stand for 10 min at room temperature. The supernatant was then adjusted to the optical density $OD_{630nm}$ = ~0.095, a turbidity matching the 0.5 McFarland standard ($1.5 \times 10^8$ colony forming units CFU/mL).

2.5.3. Antibacterial Assay

In vitro antibacterial studies were carried out by the broth microdilution method, as described previously [22]. Briefly, 15 μL ($2.25 \times 10^6$ CFU) of the inoculum was inoculated into each well of a flat 96-well microplate (Costar, Corning, NY, USA), containing 200 μL of different concentrations of the extracts (3.125–400 μg/mL) in Mueller Hinton Broth (MCD Lab; Mexico). Organic extracts were first dissolved in dimethyl sulfoxide (DMSO) and subsequently diluted in sterile broth. Additionally, each antibacterial test included wells containing the culture media plus DMSO (2%), to obtain a control measure of the solvent's antibacterial effect. Gentamicin (12 μg/mL) (AMSA; Mexico City, Mexico) was used as positive control of bacterial growth inhibition against all bacteria, except for *K. pneumoniae* BLEE+ and *E. coli* BLEE+, for which the antibiotic meropenem (12 μg/mL) (Laboratorios

Química Son's S.A. de C.V., Puebla, Mexico) was used. Bacterial cultures were incubated at 37 °C for 48 h. Plates were read at 630 nm in an enzyme-linked immunoassay (ELISA) microplate reader (Benchmark Microplate Reader; Bio-Rad, Hercules, CA, USA) at 0, 12, 24, and 48 h. The optical density ($OD_{630nm}$) was corrected by subtracting the $OD_{630nm}$ from wells with compound alone in sterile broth. The minimal inhibitory concentration was defined as the lowest compound concentration that inhibited at least 50% ($MIC_{50}$) or 90% ($MIC_{90}$) of the bacterial growth after incubation at 37 °C for 24 h. MICs were determined using the following criteria [23]:

$$MIC_{50}: (OD_{630nm} \text{ untreated bacteria} - OD_{630nm} \text{ test concentration})/(OD_{630nm} \text{ untreated bacteria}) \times 100 \geq 50\%$$

$$MIC_{90}: (OD_{630nm} \text{ untreated bacteria} - OD_{630nm} \text{ test concentration})/(OD_{630nm} \text{ untreated bacteria}) \times 100 \geq 90\%$$

### 2.5.4. Statistical Analysis

Antibacterial results are expressed as mean ± standard deviation of three independent experiments. Statistical analysis performed was one-way analysis of variance (Tukey), and the graphs of bacterial growth kinetics were performed with GraphPad Prism© Version 5.01 software.

### 2.6. Toxicity Assays

### 2.6.1. Animals

Male CD1 mice (25 to 30 g) were used for this study. The animals were fed with Rodent Lab Chow 5001 (Agribrands, Purina, Mexico) and water ad libitum. Animals were fasted for 16 h prior to their use for the assays. Animal care and experimental procedures were carried out according to the Mexican Official Standard (NOM 062 Z00 1999) for the use and care of laboratory animals, which is in accordance with the European Community guidelines (EEC Directive of 1986; 86/609/EEC).

### 2.6.2. Acute One Dose Assay

A unique 100 mg/kg dose of *S. buxifolium* and *P. gymnospora,* was administered to 5 mice using an oral cannula (Animal feeding needles, 20G, X1 11/2" Poper and Sons, Inc. Newhyde Park, NJ, USA). A control group of 5 received only saline solution (0.2 mL). The mice were observed daily over 7 days to identify any behavioral or clinical manifestations of oral acute toxicity, such as diarrhea, salivation, irritability, seizures, ataxia, and death.

### 2.6.3. Acute Three Doses Assay

A group of 5 mice received a dose of 50 mg/kg of the *S. buxifolium* and *P. gymnospora* hexanic extracts, 3 times with a 1 h time interval—150 mg/kg total. A control group of 5 received only saline solution (0.2 mL). The mice were observed daily over 7 days to identify any behavioral or clinical manifestations of oral acute toxicity, such as diarrhea, salivation, irritability, seizures, ataxia, and death.

### 2.6.4. Subchronic Assay

A group of 5 mice received a daily dose of 50 mg/kg of the *S. buxifolium* and *P. gymnospora* hexanic extracts over a 14-day period. A control group of 5 received only saline solution (0.2 mL) for the same period of time. The last day, animals were weighed and sacrificed; the kidneys, heart, and liver were extracted and weighed immediately on the analytical balance.

### 2.6.5. Acute Toxicity Testing ($LD_{50}$) Using the Lorke Method

To determine the degree of toxicity of *S. buxifolium* and *P. gymnospora* at high doses, the Lorke method was implemented [24]. Nine mice distributed in groups of three animals received 10, 100, and 1000 mg/kg of *S. buxifolium* and *P. gymnospora* once, administered by oral cannula. Observations and weighing were carried on for 14 days.

### 2.6.6. Statistical Analysis

Means and standard deviations (SD) were calculated using Excel (Microsoft Office, 2019). Statistical analysis of differences was carried out by analysis of variance (ANOVA) using SPSS 10.0 for Windows (Microsoft, Redmond, WA, USA). A $p$ value < 0.05 (Student's $t$ test) was considered significant. In all cases, the data represent three independent experiments performed in triplicate.

### 3. Results

The maceration processes of *S. buxifolium* and *P. gymnospora* seaweeds in hexane showed yields of 11.31% and 10.25%, respectively, after solvent evaporation; the ethyl acetate, methanol, ethanol, and aqueous had lesser yields and all were mainly mannitol, (Figures S3 and S4) that is the reason this study focuses on the hexane fraction.

Total hexanic extracts of *S. buxifolium* and *P. gymnospora* were analyzed with [1]H-NMR and GC (Figures S1 and S2). Figure 1 showed the [1]H-NMR spectra of the hexanic extracts of *S. buxifolium* (1) and *P. gymnospora* (2). These fatty acids included polyunsaturated ones such as ω-3; in addition, the characteristic signals of glycerol around 4 ppm were not observed, which suggests that these fatty acids were not esterified.

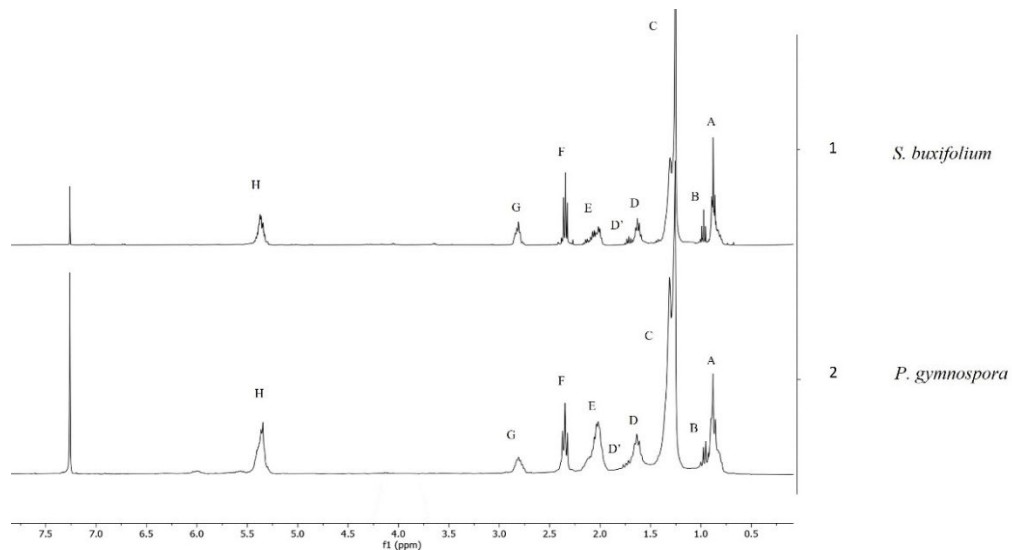

**Figure 1.** [1]H-NMR spectra of *S. buxifolium* and *P. gymnospora* hexanic extracts.

At 0.87 and 0.96 ppm, a multiplet (signal A) and a triplet (signal B) corresponding to the FA methyls and the ω-3 FA methyls protons, respectively, were observed. The C multiplet displaced at 1.27 ppm corresponded to the methylenes protons in the β position, or to the double bonds, or to the methylenes in the γ position, or to the carbonyl group. Signals D and F correspond to the protons of the methylene in position β and the α carbon in the carbonyl group, respectively. The D' multiplet represents the β methylene protons or the carbonyl of EPA (eicosapentaenoic acid). The multiplet G corresponds to the bisallylic methylene protons. Finally, the H multiplet at 5.37 ppm corresponds to the protons that form the unsaturations in the FA chains. In Table 1 are described the chemical shifts previously mentioned.

Table 2 shows the methyl esters of the fatty acids identified in the hexanic extracts of *S. buxifolium* and *P. gymnospora.* In the *S. buxifolium* hexanic extract, methyl palmitate (35.5%) was the one found in the highest proportion, followed by methyl palmitoleate (16.4%). Other methyl esters that were found were methyl oleate, stearate, linoleate, and linolenate in lower proportions. Meanwhile, the hexanic extract of *P. gymnospora* showed a higher proportion of oleic acid methyl ester (40.2%), followed by methyl palmitate (18.5%). Palmitoleate, linoleate, linolenate, and methyl stearate were found in lower proportions.

**Table 1.** FA characteristic chemical shift values observed in the $^1$H-NMR spectrum.

| Signal | Chemical Shift (ppm) | Proton Type |
|---|---|---|
| A | 0.83–0.92 (m) | Terminal-$CH_3$ group of all FA (exception ω-3 FA) |
| B | 0.97 (t) | Terminal-$CH_3$ group of unsaturated ω-3 FA |
| C | 1.17–1.40 (m) | -$(CH_2)_n$- group protons of FA chains |
| D | 1.56–1.66 (m) | Acyl-OCO-$CH_2$-$CH_2$- group protons of the beta position to carbonyl group |
| D′ | 1.70 (m) | Acyl-OCO-$CH_2$-$CH_2$ group protons of the beta position to carbonyl group of EPA |
| E | 1.94–2.12 (m) | -$CH_2$-CH=CH-$CH_2$- group protons in alpha position to double bond |
| F | 2.31 (t) | -OCO-$CH_2$- group protons in alpha position to carbonyl group |
| G | 2.72–2.90 (m) | -CH=CH-$CH_2$-CH=CH- group protons of polyunsaturated ω-6 and ω-3 acyl groups and FA |
| H | 5.30–5.42 (m) | -$CH$=$CH$- vinylic protons of FA chains |

**Table 2.** Identification of FAMEs in hexanic extracts of *S. buxifolium* and *P. gymnospora*.

| Name | *S. buxifolium* | | *P. gymnospora* | |
|---|---|---|---|---|
| | Retention Time (min) | % Area | Retention Time (min) | % Area |
| Methyl palmitate | 4.67 | 35.5 | 4.67 | 18.5 |
| Methyl palmitoleate | 5.07 | 16.4 | 5.07 | 8.1 |
| Methyl stearate | 8.74 | 3.3 | 8.74 | 1.1 |
| Methyl oleate | 9.31 | 7.5 | 9.32 | 40.2 |
| Methyl linoleate | 10.81 | 1.6 | 10.81 | 5.6 |
| Methyl linolenate | 13.35 | 1.1 | 13.36 | 1.7 |

Both hexanic extracts contained the most common fatty acids, though with differences in proportions. The hexane extract of *S. buxifolium* was characterized by having an unsaturated FA with 16 carbons in high abundance, whereas the hexane extract of *P. gymnospora* was characterized by having an 18-carbon unsaturated FA, ω-9, with high abundance.

### 3.1. Biological Activity

### 3.1.1. Antibacterial

Broth microdilution method was used to evaluate the inhibitory activity against nine bacterial strains. *S. buxifolium* hexanic extract was the most active against Gram-positive (*E. faecalis*, *S aureus*, and *S. epidermidis* with $MIC_{50}$ of 25, 200, and 200 µg/mL, respectively; and $MIC_{90}$ of 400 µg/mL for *S aureus* and *S. epidermidis*) and Gram-negative bacteria (sensitive and ESBL + *K. pneumoniae* exerting a $MIC_{50}$ of 400 µg/mL) (Table 3). The hexanic extract of *P. gymnospora* showed antibacterial activity against *E. faecalis* and *S. epidermidis*: the $MIC_{50}$ was 200 µg/mL for both strains (Table 3; Figure 2).

**Table 3.** Growth-inhibitory activity of algae extracts or compounds against different Gram-positive and Gram-negative bacteria. * Concentration in µg/mL; [a]: resistant bacteria; —: no $MIC_{50}$ or $MIC_{90}$ reached at 400 µg/mL; ESBL+: extended spectrum beta-lactamase.

| Bacteria | Strain | *P. gymnospora* | | *S. buxifolium* | |
|---|---|---|---|---|---|
| | | $MIC_{50}$* | $MIC_{90}$* | $MIC_{50}$* | $MIC_{90}$* |
| Gram-positive bacteria | *Enterococcus faecalis* ATCC 51299 | 200 | — | 25 | — |
| | *Staphylococcus aureus* ATCC 25293 | — | — | 200 | 400 |
| | *Staphylococcus epidermidis* | 200 | — | 200 | 400 |
| Gram-negative bacteria | *Escherichia coli* ATCC 25292 | — | — | — | — |
| | *Klebsiella pneumoniae* | — | — | 400 | — |
| | *Pseudomonas aeruginosa* | — | — | — | — |
| | *Salmonella typhimurium* | — | — | — | — |
| | [a] *Escherichia coli* ESBL+ | — | — | — | — |
| | [a] *Klebsiella pneumoniae* ESBL+ ATCC 700603 | — | — | 400 | — |

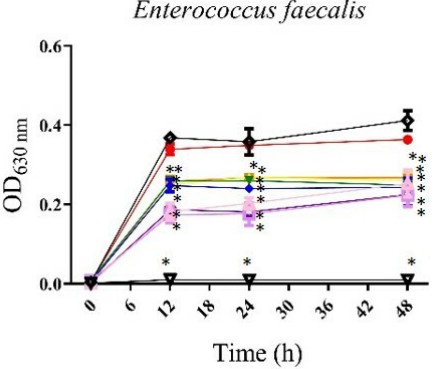

**Figure 2.** Antibacterial activity of *P. gymnospora* extract against *E. faecalis* and *S. epidermidis* evaluated at 3.125–400 μg/mL. Gentamicin was used as the positive control. (●, 3.125 μg/mL; ■, 6.25 μg/mL; ▲, 12.5 μg/mL; ▼, 25 μg/mL; ◆, 50 μg/mL; ⬭, 100 μg/mL; □, 200 μg/mL; △, 400 μg/mL; ▽, gentamicin 12 μg/mL; ◇, bacteria). All values represent mean of triplicate determinations ± SD. Significant differences ($p < 0.05$) from bacterial growth control are marked with asterisks.

None of the extracts were active against the Gram-negative bacteria *E. coli*, *P. aeruginosa*, *S. typhimurium*, and *E. coli* ESBL +. Neither *S. buxifolium* nor *P. gymnospora* extracts exerted greater antibacterial activity than the positive controls (gentamicin and meropenem) (Figures 2 and 3).

The best MIC$_{50}$ (25 μg/mL) occurred for *E. faecalis* by *S. buxifolium* extract, and an antistaphylococcical effect of this extract was revealed by the MIC$_{90}$ reached at 400 μg/mL (Figure 3). Finally, it is important to highlight the activity of *S. buxifolium* against the resistant *K. pneumoniae* ESBL+ and the selectivity of *P. gymnospora* against Gram-positive bacteria.

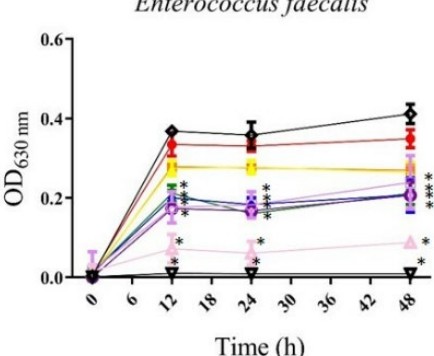
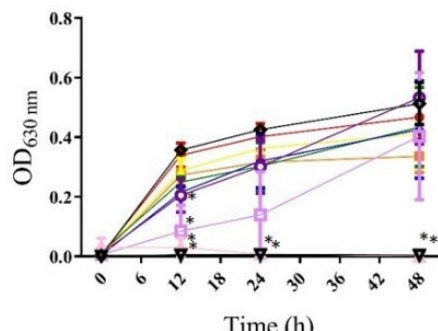

**Figure 3.** *Cont.*

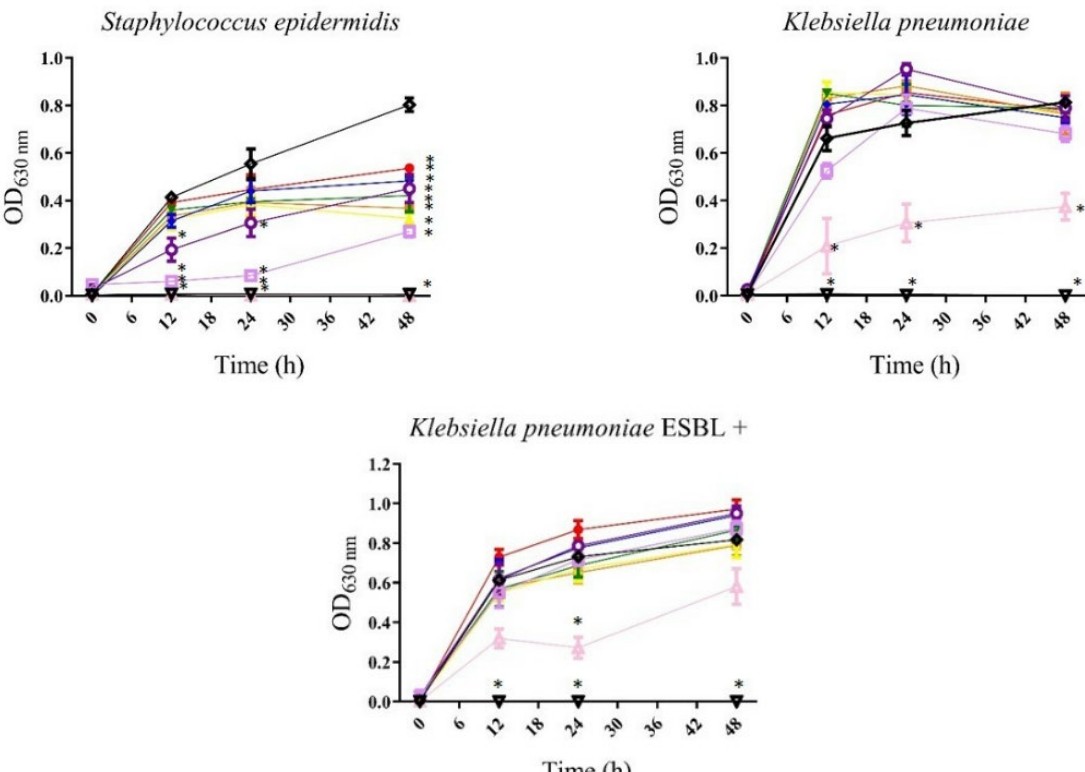

**Figure 3.** Antibacterial activity of *S. buxifolium* extract against Gram-positive and Gram-negative bacteria evaluated at 3.125–400 µg/mL. Gentamicin was used as positive control for all bacteria except for *K. pneumoniae* ESBL+, for which meropenem was used (●, 3.125 µg/mL; ■, 6.25 µg/mL; ▲, 12.5 µg/mL; ▼, 25 µg/mL; ♦, 50 µg/mL; O, 100 µg/mL; □, 200 µg/mL; △, 400 µg/mL; ▽, gentamicin or meropenem 12 µg/mL; ◇, bacteria). All values represent mean of triplicate determinations ± SD. Significant differences (*p* < 0.05) from bacterial growth control are marked with asterisks.

### 3.1.2. α-Glucosidase Inhibitory Activity of Hexanic Extracts

To evaluate the potential of hexanic extracts of *S. buxifolium* and *P. gymnospora* for diabetes treatment, the α-glucosidase inhibitory activity of it was evaluated. Preliminary results showed that the hexanic extract of *S. buxifolium* was more active than that of *P. gymnospora* at 1, 10, and 100 µg/mL (Figure S6). The $IC_{50}$ value of *S. buxifolium* was 36.9 µg/mL (Figure 4).

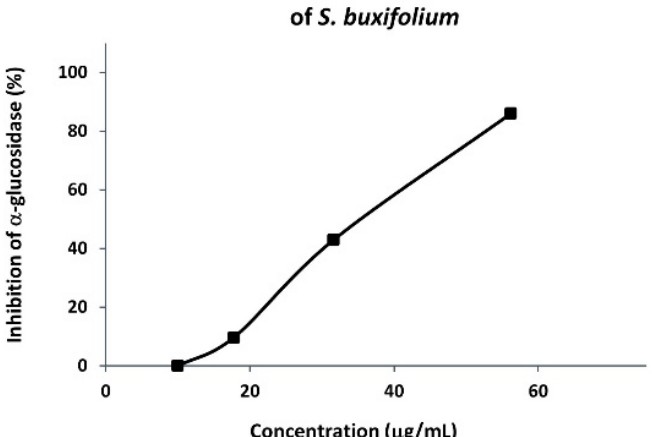

**Figure 4.** α-Glucosidase inhibitory activity of the hexanic extract of *S. buxifolium*.

*3.2. Toxicity of S. buxifolium and P. gymnospora Hexanic Extracts*

In the acute one-dose assay, treated animals showed somnolence in comparison to control animals, after 15 min of administration and normalization for 50 min. At the end of the assay, treated animals gained weight in relation to the control group. In the acute three-dose assay (Figure S5), no signs of toxicity were observed.

In the subchronic assay, the animals showed no signs of damage or statistical differences in the relative weight of the organs, which suggests the daily administration of *S. buxifolium* and *P. gymnospora* in the treated group induced microsomal activity in the liver that facilitated the excretion of the *S. buxifolium* and *P. gymnospora* or of their metabolites so that they did not accumulate it and did not show any effect on the weight of the treated animals.

In the test to determine the $LD_{50}$, at the end of the assay (14 days), no deaths were recorded, even at doses as high as 1000 mg/kg, which suggests *S. buxifolium* and *P. gymnospora* have no toxicity at those doses tested, in the model used.

Table S1 indicates that there was no mortality, and the hexanic extracts did not exhibit related signs and symptoms of toxicity in 7 and 14 days.

## 4. Discussion

Previous studies indicated the roles and biological activities of mannitol and polar extracts [25] for different species of *Sargassum* [26] and *Padina* [27]. However, there is little research focused on the use and biological activity of the non-polar fractions of algae.

One of the most common ways to identify fatty acids is through GC, with prior derivatization forming (FAMEs). FAMEs are more volatile and can be separated by GC identified by retention times compared to standards [28,29].

The variation in the FA profile between different species of *Sargassum* was expected due to it having been reported, which opens the potential of using the content of fatty acids as a chemotaxonomic tool [30]. Both extracts (Figure 1) showed a mixture of characteristic fatty acids in marine species [31,32] and vegetable oils [33,34]. Saturated (palmitic acid) and unsaturated (oleic acid) fatty acids were the main compounds in the hexanic extracts, which are valuable for their biological activity and as nutrients [35,36]. [1]H-NMR spectra showed chemical shifts of the characteristic proton signals that have been observed in mixtures of fatty acids and that have been described in various investigations. In addition, a characteristic signal of EPA in [1]H-NMR spectra was observed, which is a fatty acid in high demand for its health benefits [37].

Regarding the in vitro antimicrobial experiments, some antibacterial compounds have been described previously from brown seaweeds: the phytosterol saringosterol isolated from *Lessonia nigrescens* inhibited *Mycobacterium tuberculosis* H37Rv with a MIC of 0.25 µg/mL [38]; phlorotannins from *Ecklonia kurome* showed bactericidal activity against food-borne pathogenic bacteria, methicillin-resistant *S aureus* (MRSA), and *Streptococcus pyogenes* [39]; diterpenes featuring the dolabellane skeleton were isolated from *Dilophus spiralis* and exerted inhibitory activity against six *S aureus* strains with MICs ranging from 2 to 128 µg/mL [40]; spiralisones and chromones from *Zonaria spiralis* displayed inhibitory activity against *Bacillus subtilis* [41]; and phlorofucofuroeckol-A (PFF) isolated from *Eisenia bicyclis* was active against MRSA, having a synergistic effect with the β-lactam antibiotics ampicillin, penicillin, and oxacillin [42].

Moreover, a novel chromene isolated from Homoeostrichus formosana inhibited the growth of S. typhimurium and Yersinia enterocolitica [43]; fucofuroeckol-A from E. bicyclis exhibited anti-*Listeria monocytogenes* potential and synergy with streptomycin [44]; fucoidans from Fucus vesiculosus inhibited the growth of all microorganisms tested, showing a bacteriostatic effect and MICs in the range of 4 to 6 mg/mL [45].

Regarding the antibacterial mechanisms of action of algal compounds, some research groups have shed light about them: dieckol, a naturally occurring phlorotannin found in some brown algal species, possesses antibacterial effects due to cell-wall destabilization, rupture of the peptidoglycan layer, osmotic imbalance, release of intracellular components,

and inhibition of molecular processes such as DNA replication, transcription, translation, and enzyme production, leading to bacterial death [46]; PFF isolated from *E. bicyclis* significantly suppressed in SARM the expression of the methicillin resistance-associated genes and the production of penicillin-binding protein 2a (PBP2a) [47]; depolymerized fucoidans from *Laminaria japonica* were bactericidal to *S aureus* and *E. coli* by destruction of the cytomembranes and targeting membrane proteins [48]. Finally, dolastane diterpenes from *Canistrocarpus cervicornis* modulated the drug resistance in *S aureus*, acting as antibiotic adjuvants and as potential inhibitors of efflux pump [49], a mechanism that could be explored in order to explain the antistaphylococcical effect shown by both seaweeds of our research.

The free fatty acids (FFA) found in the hexanic extracts of *S. buxifolium* and *P. gymnospora* could be contributing to the antibacterial activity by producing disruption of the electron transport chain (ETC) and oxidative phosphorylation, inhibiting the FA biosynthesis, and/or inducing the fugue of bacterial metabolites by pore formation [50,51]; however, further antimicrobial experiments are needed to prove the hypothesis.

*S. buxifolium* and *P. gymnospora* extracts were not able to inhibit the growth of *P. aeruginosa*; however, in contrast with our results, other brown algae dichloromethane and ethyl acetate extracts from *Stypocaulon scoparium* were demonstrated to inhibit *P. aeruginosa* biofilm formation [52]; this difference could be explained by the polarity of the main compounds extracted.

Other *Sargassum* and *Padina* species have been reported previously to be antibacterials. *Sargassum latifolium, Sargassum platycarpum*, and *Sargassum tenerrimum* were active against Gram-positive and Gram-negative bacteria [53,54]; and *Sargassum macrocarpum* yielded sargafuran active against *Propionibacterium acnes* [55], just to mention some examples.

Extracts of *Padina sanctae-cruces* combined with drugs of the class of aminoglycosides were synergistic against *E. coli* [56]; and organic algal extracts of Padina sp. presented inhibitory activity against *B. cereus* (MIC 63 μg/mL) and *S aureus* (MIC 130 μg/mL); however, they were inactive against the Gram-negative bacteria used [57], which is in agreement with our results: the *P. gymnospora* hexanic extract was not active against the Gram-negative bacteria. This could be explained by the lipophilicity of the compounds extracted by n-hexane, as the Gram-negative bacterial membrane contains lipopolysaccharides that create a hydrophilic barrier that may prevent the entry of the low-polarity compounds [57].

Specifically, *P. gymnospora* was previously studied against many human pathogens and fungal strains by using the disc diffusion method [58]; however, to the best of our knowledge, this is the first report of *S. buxifolium* having antibacterial properties.

Finally, this is the first report of *S. buxifolium* activity against *K. pneumoniae* ESBL+. Responsible chemical compounds and possible mechanisms of action remain to be studied; however, undoubtedly, these brown algae are a potential source of novel antimicrobial compounds against sensitive and resistant bacteria.

Only a few studies have focused on the extraction and characterization of the chemical compounds and metabolites of these algae [59,60]

The hexanic extract of *S. buxifolium* had the best a-glucosidase inhibition compared with *P. gymnospora*, (Figure S6). The $IC_{50}$ of the algae was 36.9 μg/mL. This research reaffirms that the *Sargassum* genus has $\alpha$-glucosidase inhibitory activity [61]. It also brings new information about the hexanic fraction in the genus and for the species *S. buxifolium*, which had not been taken into account [60]. Other investigations about $\alpha$-glucosidase inhibition of diverse brown seaweed showed that palmitic acid was one of the most abundant components and is considered a potential $\alpha$-glucosidase inhibitor [62]. This fact suggests that the $\alpha$-glucosidase inhibitory activity of *S. buxifolium* hexanic extract was due to the presence of this FA, which was the majority in the extract, according to the GC analysis. This FA was in lesser proportion in the hexanic extract of *P. gymnospora*, showing less inhibitory activity of the $\alpha$-glucosidase enzyme of this extract.

Acute toxicity study gives information about $LD_{50}$, therapeutic index, and the degree of safety of a pharmacological new agent [63]. Chronic treatment in this study showed that both extracts were well tolerated by all animals.

## 5. Conclusions

The present work focused on the determination of the potential antidiabetic, antimicrobial, and toxicity profiles of hexanic extracts of *S. buxifolium* and *P. gymnospora*. The results of this work suggest that the algae contain substances that are capable of inhibiting the growth of resistant bacteria and have antidiabetic activity. These extracts can be classified as nontoxic, and our research suggests they may contribute to the development of new treatments.

**Supplementary Materials:** The following supporting information can be downloaded at: https://www.mdpi.com/article/10.3390/ijpb14010006/s1. Figure S1: Chromatogram, FAMEs from hexanic extract of *Sargassum buxifolium*. Figure S2: Chromatogram of FAMEs from hexanic extract of *Padina gymnospora*. Figure S3: $^1$H-NMR spectrum of mannitol (DMSO-d6, 400 MHz). Figure S4: $^{13}$C-NMR spectrum of mannitol (DMSO-d6, 100 MHz). Figure S5: Toxicity. Figure S6: α-glucosidase inhibitory activity of hexanic extracts of *P. gymnospora* and *S. buxifolium*. Table S1: Effects of the acute oral treatment with *S. buxifolium* and *P. gymnospora*.

**Author Contributions:** J.J.A.-S. designed the study and performed the research. M.J.-E. and A.G.A.-O. were involved in the study design, organization, resourcing, and writing of the paper together with all other authors. R.T.-H. designed and performed the chemical analysis. S.D.-M. and L.H.-A. designed and performed collection and taxonomic identification. E.W.C.-A. designed and performed bacterial assays. C.I.E.-P. and R.R.-C. supervision and methodology applications. All authors have read and agreed to the published version of the manuscript.

**Funding:** This research was funded by UNAM-PAPIIT, grant number BG200321, IN225416, IA204921, IN222721.

**Institutional Review Board Statement:** "Ethical review and approval were waived for this study due to REASON (please provide a detailed justification)."

**Informed Consent Statement:** Not applicable.

**Data Availability Statement:** Not applicable.

**Acknowledgments:** Jesus Javier Alvarado Sansininea acknowledges the fellowship "Estancias posdoctorales DGAPA/UNAM". R.T.H received a fellowship from UNAM-PAPIIT, B221296. The authors are grateful to Instituto de Química, UNAM; Antonio Nieto Camacho, María Teresa Ramírez, Lucía del Carmen Márquez, Lucero Mayra Ríos Ruíz, Eréndira García, Beatriz Quiroz García, Elizabeth Huerta Salazar, María de los Ángeles Peña González, and M. en C. Blanca Verónica Juárez Jaimes for technical assistance; and Nathaly Montoya-Camacho and Martin Samuel Hernández-Zazueta for technical assistance in the antibacterial assay.

**Conflicts of Interest:** The authors declare no conflict of interest.

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
