# Peer review of "Antibacterial, Antidiabetic, and Toxicity Effects of Two Brown Algae: Sargassum buxifolium and Padina gymnospora"

_2037-0164, doi:10.3390/ijpb14010006_

Round 1

Reviewer 1 Report

The paper reports the antibacterial, antidiabetic and toxicity effects of two Brown algae: Sargassum buxifolium and Padina gymnospora. The topic seems worthy of investigation and the paper is within the journal scoop. The experiments are well designed, and the conclusions were confirmed by the obtained results, please consider the following comments:

In the title, please delete “evaluation of”.

Please explain: 1 mL of 0.5M KOH was added to 16.2 86 mg of the hexane extract of S. buxifolium. Did you extracted the fatty acids using Soxhlet method or the hexane extract was concentrated to be used for FAMEs analysis!!? This information should be explained.

Figure 1 replace it by a high-quality figure and delete red underline for the Latin names. Add to the fig. footnote What is the significance of letters A to H in the chromatogram

The table title should appear above the table.

Table 2 : same suggestion the table title should appear above the table,  and please change the retention time to 2 digits below zero: ex. 4.668 should be 4.69

Figures 3 and 4 should be replaced by high quality figures.

Figures 5 and6 are confusing, the related titles are not correct, both could be placed in one graph and the control is missed please ad acarbose or other positive control to evaluate the remaining inhibitory activity.

Line308 “mg / kg » there is no space before and after /. However please make sure that there is a space between numbers and units. Check the whole manuscript.

The references should be uniform and prepared according to the journal style.

Some illustrations could be placed into the supplementary file. (ex. Table 4)

Author Response

Hi.

Reviewer 2 Report

The authors reported the phytochemical components and their antibacterial, antidiabetic and toxic potential of the hexanic fraction of two Brown algae: Sargassum buxifolium and Padina gymnospora. The results are interesting and useful to develop new antimicrobial agents from Brown algae. Before accepting and publishing the article, the following questions need to be revised and/or clarified.

1.       The authors should check Figure 5 and Figure 6 carefully. Preliminary results showed that hexanic extract of S. buxifolium was more active than P. gymnospora (Figure 5) at 1, 10 and 100 mg/mL. The IC50 value of P. gymnospora was of 36. 9 mg/mL (Figure 6).

2.       Many writing/grammar errors need to be corrected. Such as CH2, CH3 (Table 1), MIC50, MIC90 (Table 3), IC50, CDCl3, ……, subscript. a-Glucosidase, α-Glucosidase, ……

3.       The fatty acid profile was determined by HPLC or GC?

Author Response

Hi.

Round 2

Reviewer 1 Report

the paper was revised carefully according to the previous comments. The current version of the manuscript is within the standard and can be considered for publication. I have no further comment.